# Cross-Reactivity of Intraoral Allergic Contact Mucositis in the Nickel-Sensitized Ear Model of Metal Allergy

**DOI:** 10.3390/ijms24043965

**Published:** 2023-02-16

**Authors:** Ryota Matsubara, Kenichi Kumagai, Keisuke Nasu, Takamasa Yoshizawa, Kazutaka Kitaura, Motoaki Suzuki, Yoshiki Hamada, Ryuji Suzuki

**Affiliations:** 1Department of Oral and Maxillofacial Surgery, Sendai Tokushukai Hospital, Sendai 981-3116, Japan; 2Department of Rheumatology and Clinical Immunology, Clinical Research Center for Rheumatology and Allergy, Sagamihara National Hospital, National Hospital Organization, Sagamihara 252-0392, Japan; 3Department of Oral and Maxillofacial Surgery, Dentistry and Orthodontics, The University of Tokyo Hospital, Tokyo 113-8655, Japan; 4Department of Oral and Maxillofacial Surgery, School of Dental Medicine, Tsurumi University, Yokohama 230-8501, Japan; 5Repertoire Genesis Inc., Osaka 567-0085, Japan; 6Department of Anatomy and Physiology, Faculty of Medicine, Saga University, Saga 849-8501, Japan

**Keywords:** metal allergy, nickel, palladium, chromium, cross-reactivity, allergic contact mucositis, metal-specific T cells

## Abstract

Cross-reactivity of metal allergies can make metal allergy treatment complicated because the background of immune response in cross-reactions remains unknown. In clinical settings, cross-reactivity among several metals has been suspected. However, the precise mechanism of immune response in cross-reactivity is unclear. Two sensitizations with nickel, palladium, and chromium plus lipopolysaccharide solution into the postauricular skin were followed by a single nickel, palladium, and chromium challenge of the oral mucosa to generate the intraoral metal contact allergy mouse model. Results showed that the infiltrating T cells in nickel-sensitized, palladium- or chromium-challenged mice expressed CD8+ cells, cytotoxic granules, and inflammation-related cytokines. Thus, nickel ear sensitization can cause cross-reactive intraoral metal allergy.

## 1. Introduction

Metal allergy is a delayed-type hypersensitivity reaction in which T cell-dependent macrophage activation and inflammation cause tissue injury [1]. Unlike immediate hypersensitivity reactions, cellular hypersensitivity reactions, such as delayed-type hypersensitivity, are mediated by antigen-specific effector T cells. The liquefaction of metal materials causes metal hypersensitivity or allergic reactions that mediate antigen-specific T cell sensitization.

Previous studies aimed to evaluate antigen-specific immune mechanisms by developing mouse models of palladium (Pd), nickel (Ni), chromium (Cr), and titanium (Ti) allergies by sensitizing the skin with chloride and lipopolysaccharide (LPS) solutions and by challenging it with the injection of these metal solutions into the footpads and oral mucosa [2,3,4,5,6].

Previous data showed a heterogeneous group of patients with different manifestations of oral contact allergy to dental metals [7]. Generally, Pd-ACD is almost observed together with Ni-ACD [8,9]. Antibodies against one antigen may bind with other structurally similar antigens. Such binding to similar epitopes is referred to as cross-reaction [1]. This phenomenon may be mainly attributed to cross-reactivity between Pd and Ni. However, the precise mechanism is not completely understood [10,11,12,13]. Therefore, a novel mouse model of cross-reactive metal allergy in the footpad skin was established, and immune response was investigated [14]. However, the mucosal immune system has a different immune response than the skin immune response because it encounters antigens more frequently and more extensively [15,16].

Ni is the most common metal causing contact dermatitis [17]. It is found in several personal products, such as ear piercing, jewelry, belt buckles, metal fasteners on clothing, and eyeglass frames [18]. Among them, ear piercing is a sensitizer for developing Ni allergies [19]. However, whether allergies to other metals are involved is unknown.

Adverse reactions to metal ions, such as cheilitis, perioral dermatitis, burning mouth syndrome, lichenoid reaction, orofacial granulomatosis, pustulosis palmaris et plantaris, rheumatoid arthritis, and systemic lupus erythematosus, can cause serious issues due to incompatibility reactions to metal-containing biomaterials [20,21,22,23]. Nevertheless, the precise mechanism of cross-reactivity among metal allergens in the mucosa remains unknown. To elucidate the immune response of cross-reactive Ni allergy, the current study aimed to establish a novel mouse model of Ni ear sensitization and characterize intraoral-infiltrating T cells during the elicitation phase in terms of phenotypic T cell markers and cytokine expressions.

## 2. Results

### 2.1. Oral Mucosa Swelling in a Metal Allergy Cross-Reaction Mouse Model

All experimental protocols are depicted in the Materials and Methods section (Table 1). In all groups, the peak of buccal mucosa swelling was observed at 1 day after challenge. At 7 days after challenge, buccal mucosa swelling was significantly higher in the Ni-Pd and Ni-Cr groups than that in the control group. Meanwhile, swelling did not significantly differ between Pd-Ni and Cr-Ni mice and control mice. Buccal mucosa swelling was significantly higher in sensitization of Ni-induced allergic mice compared with sensitization of Pd-induced allergic mice (Figure 1). Additionally, visually significant swelling was observed in the buccal area of mice with Ni-Pd and Ni-Cr allergic contact mucositis (ACM) compared with control mice at day seven after the first challenge (Figure 2).

### 2.2. Histological and Immunohistochemical Analyses of CD3 and F4/80 in the Oral Mucosa of Mice with Cross-Reactive Metal-Induced Allergy

To validate whether antigen-presenting cells (APCs) and T cells infiltrated into the site of inflamed skin, we analyzed the oral mucosa of metal-induced ACM and control mice at 1 and 7 days after the challenge. Hematoxylin and eosin (H&E) staining showed epithelial acanthosis and epidermal spongiosis and liquefaction degeneration of the epithelial basal layer infiltrated with dense mononuclear cells in the epithelial basal layer and upper dermis of ACM mice (Figure 3 Day1-C, Day7-B,C). Immunohistochemical staining showed that CD3-positive T cells existed in the epithelial basal layer and upper dermis of ACM mice (Figure 3 Day1-F,G, Day7-F,G). Immunohistochemical staining revealed that F4/80-positive cells predominantly existed in the epithelial basal layer and upper dermis of ACM mice (Figure 3 Day1-J,K, Day7-J–L). In the Pd-Ni groups, F4/80 was present only after 7 days. In contrast, inflammatory reactions (H&E, CD3-positive T cells, F4/80-positive cells) were not observed in the oral mucosa of the control mice (Figure 3).

### 2.3. Expression Levels of T Cell Markers, Related Cytokines, and APC-Related Markers in the Oral Mucosa of Mice with Cross-Reactive Metal-Induced Allergy

We investigated the expression levels of T cell markers, related cytokines, and APC-related markers via quantitative polymerase chain reaction (qPCR). Messenger RNA (mRNA) expression levels of CD4 and CD8 in the left and right buccal mucosa were assessed at 1 and 7 days after the challenge. To validate whether T cells infiltrated into the inflamed oral mucosa in ACM mice, we performed qPCR analysis of CD4 and CD8 expressions. Metal-induced ACM mice had significantly higher CD8 levels than control mice 1 and 7 days after the challenge. The CD8 levels of Pd-Ni group and Ni-sensitized mice (Ni-Pd, Ni-Cr) were significantly higher than control mice at 1 day after challenge, and Ni mice (Ni-Pd, Ni-Cr) were significantly higher than control mice even at 7 days after challenge (Figure 4).

Further, we examined the expression levels of Th1 cytokines (tumor necrosis factor [TNF]-α and interferon [IFN]-γ), Th2 cytokines (IL-4 and IL-5), cytotoxic granules (granzyme A and B), transcription factors of regulatory T cells (Foxp3), CD1d-restricted T cells (CD-1d), and MHC-related protein 1 (MR1) in the left and right buccal mucosa at 7 days after challenge (Figure 5). The levels of IFN-γ, IL-4 and granzyme B levels were significantly higher in sensitization of Ni mice (Ni-Pd, Ni-Cr) than control mice at 7 days after challenge (Figure 5A–C). The levels of TNF-α, granzyme A, CD-1d, and MR1 levels were significantly higher in Pd-Ni groups and sensitization of Ni mice (Ni-Pd, Ni-Cr) than control mice at 7 days after challenge (Figure 5A,B,D).

## 3. Discussion

Delayed type hypersensitivity generally occurs when the swelling takes 24 to 48 h after challenge, and onset of the pathology may take three to five days [24]. Clinically, the patch test is evaluated until 1 week later. In this study, in all groups, the peak of buccal mucosa swelling was observed at 1 day after challenge. Moreover, allergic reactions were most observed at 7 days after challenge. Infiltrating T cells in the Ni-sensitized, Pd- and Cr-challenged mice expressed the most CD8+ cells 7 days after the challenge. However, infiltrating T cells in Pd- and Cr-sensitized, Ni-challenged mice did not express CD8+. Ni-sensitized mice had significantly higher expression levels of almost all immune response-related genes. Cr-sensitized, Ni-challenged mice almost did not show significant differences in buccal mucosal swelling or in the expression levels of immune response-related genes. Thus, we suggested that the sensitization of Ni caused significant cross-reactivity with Pd and Cr. In our previous study, the oral mucosal allergic mouse model sensitized and challenged with nickel had the highest CD8 expression at day one of elicitation for challenge [5]. It suggests that allergic reactions may differ depending on the metal exposed to challenge, even when sensitized to Ni. Cross-reactivity in allergic contact dermatitis occurs due to the similar structure of antigens. Thus, cross-reactivity is found among several foreign substances, such as food [25], antibiotics [26], antifungals [27], anti-inflammatory analgesics [28], and steroids [29]. In metals, cross-reactions between Ni and Pd are the main reactions. A previous study has shown that only Ni and Pd cross-reacted in ear skin [30]. Ni and Pd have physiochemical similarities, and they belong to the same group in the periodic table. This fact considers that cross-reactivity relates near to the periodic table of the chemical elements (PSE) [31]. Identical structures form the same complexes and cause similar modifications, and they may be recognized by the same T cells [31]. Franziska et al. reported up to 80% Pd co-sensitization with Ni [32]. However, in the oral cavity, during Ni sensitization, immune responses were observed even during Cr challenge for elicitation. This fact may be correlated with differences in the immune mechanisms of the skin and oral cavity. The skin and mucous membranes have different immune responses. Contact dermatitis is a disease that can be cured by identifying the causative allergen and discontinuing contact. However, when the cause of the disease is not clear and appropriate protective measures cannot be taken, it is often intractable and difficult to treat. Symptomatic treatment of contact dermatitis without identifying the cause is not a desirable option, with consideration of the risk of side effects due to the continuous use of topical steroids and the unnecessary expenditure of medical expenses. Treatment is mainly recommended with the removal of the causative metal and treatment with steroids or antihistamines.

In clinical settings, the treatments of metal allergy include allergen avoidance and use of antihistamine drugs and corticosteroids. However, it is challenging to determine removal because it is difficult to identify the substances causing cross-reactions [33]. The use of corticosteroids is effective in immune system diseases. However, it also has side effects. A previous study reported that antihistamines inhibited immune responses in metal allergies compared with corticosteroids [34]. However, treatment of the metal allergy intraorally are only the removal of the causative agent and topical application steroid. In addition, in rare cases, the topical steroid ointment itself can cause allergies [35]. Allergic reactions to topical medications can be caused not only by the drug but also by the base or preservative. Whether the allergy is caused by a single metal or a cross-reaction is challenging to determine due to the presence of multiple metals in a small area in the oral cavity. In such cases, it is difficult to investigate cross-reactivity. However, steroids should not be continually used if the cause is unknown. Testing methods for metal allergies include blood tests, lymphocyte transformation tests [36], oral challenge tests [37], hair mineral analyses [38], and principal component analyses [39]. Among these tests, the patch tests are the most generic test [40]. They have long been used to identify allergenic substances. Identifying allergens via patch testing can help treat refractory and recurrent allergic dermatitis. In patients with a history of metal allergy, patch testing is recommended prior to the use of metals with treatment [41]. The problem is that judgments about how to implement, judge, discuss, and guide patients’ lives based on patch test results require some practice because of the bias of the users. In addition, whether the patients have a history of immunologically relevant co-exposures is not known, and this is an issue with analyzing cross-reactivity in patch tests. Hence, the timing of patch testing is not similar, and analyzing the causality of cross-reactivity is challenging. Furthermore, some patients, including those on steroids, those who need to shave, and pregnant women, should not undergo the test. Due to the abovementioned reasons, everyone cannot be tested. However, there is still no more reliable and useful test method than the patch test in identifying the cause of the problem. Therefore, it is important to elucidate the background of immune response in metal allergy to obtain a better diagnosis.

Ni causes an allergic reaction correlated with the number of ear piercings [19]. However, the correlation between piercing and mouth allergies in Ni is unclear. It is known that contact hypersensitivity reactions to divalent cations such as Ni have been observed. These divalent cations can alter the conformation or the peptide binding of MHC class II molecules and, thus, provoke a T cell response [24]. Additionally, it is well-known that LPS is required for the development of metal allergy in mice [42]. To determine whether the challenge of metal allergy is hapten-specific, there have been reports of metal solution plus LPS sensitized mice challenged with metal solution or 1-fluoro-2,4-dinitrobenzene (DNFB) solution to induce allergic reactions [43]. Sensitized, metal ion-challenged mice showed allergic reactions, while sensitized, DNFB-challenged mice did not. DNFB-sensitized, DNFB-challenged mice exhibited an ear swelling response. However, DNFB-sensitized, Pd-challenged mice did not. Moreover, no allergic reactions were observed when unsensitized, metal solution alone sensitized, and LPS alone sensitized mice were challenged with injections of the same amount of metal. The results are known that metal allergy is hapten-specific and that a metal-specific immunological response develops in mice. Therefore, they also showed that LPS is essential for the metal allergy response. In our previous study, we used this information as the basis for an ear-sensitized mouse and challenged the oral cavity to generate a Ni allergic mouse model [5]. Results showed that mice presented with Ni allergy in the oral cavity. Therefore, we suggested that metals outside the oral cavity can be the main cause of allergies, and secondary allergic reactions may occur in the oral cavity. Thereby, in this study as well, we used LPS plus metal solution for all experimental groups in sensitization.

In the case of Ni allergy, NK T cells are involved in both the skin and oral cavity and are believed to be involved in allergic reactions [2,5]. Our study has previously suggested that natural killer (NK) T cells may be correlated with cross-reactivity in metal allergy [44]. The ability of invariant natural killer T (iNKT) cells to recognize different glycolipid constituents from microorganisms presented by CD1d molecules places them in an innate category. Meanwhile, their possession of a fully rearranged T cell receptor, despite its relatively limited repertoire, makes them adaptive [24]. Therefore, iNKT cells can be involved in innate and acquired immunity. iNKT cells acquire a defined effector program during their development in the thymus [24]. They exhibit a memory-cell phenotype when they leave the thymus and migrate to peripheral lymphoid tissues and mucosal surfaces [24]. Hence, NK T cells may act as a bridge between innate and adaptive immunity. The T cell population called mucosal-associated invariant T cells (MAIT cells) are recognized vitamin B9 metabolites presented by the MR1 MHC class Ib molecule, suggesting that the MAIT cells also have a ‘transitional’ role between innate and adaptive immunity [24,45]. Since allergy to Pd alone is rare, cross-reactivity is known to be involved in Pd allergy [46]. Lymphocyte transformation studies showed simultaneous patch test reactivity to palladium and nickel, suggesting cross-reactivity [47]. However, during Pd sensitization and Ni elicitation, there was no allergic reaction. The involvement of T cells was observed. In contrast, there were significant differences in gene expression levels of CD1d and MRI in Pd and Cr induced by Ni sensitization and Ni induced by Pd sensitization compared with controls. This study suggested a possible involvement of NK T cells and MAIT cells in the challenge of Pd and Cr by Ni sensitization and Ni challenge by Pd sensitization in the oral cavity. Therefore, in the case of Pd sensitization, innate immune-derived T cells may be involved in the development of allergy. As described in the Materials and Methods, in this mouse model, sensitization was achieved via injection into the ears and elicitation via injection into the oral cavity. Therefore, cross-reactivity between Ni and Pd in the oral cavity may be caused by Ni sensitization in the skin. Haptens can induce an early response via innate immune mechanisms [48]. Therefore, suppression of products that may cause Ni ionization may inhibit allergic reactions. Successful avoidance of this sensitization can reduce the development of allergies with complex pathologies, including cross-reactivity. In fact, Ni regulation could reduce Ni allergies. However, its use is not regulated in several countries.

This study showed cross-reactivity between Ni and other metals. This suggests that Ni suppression may inhibit allergic reactions to other metals in which Ni is an inflamer.

We consider that oral metal allergies can be controlled by properly regulating the use of metals outside the oral cavity. Therefore, a patch test even before treatment with other metals may be effective in people who are constantly exposed to Ni. In addition, since the degree of sensitization by metals varies depending on individual susceptibility, it may be important to regulate the Ni content of piercings and other products in several countries. Recently, it has been suggested that even artificial joint implants used in orthopedic surgery on human subjects be patch tested beforehand and substituted with zirconia or other alternatives, since nickel can be allergenic [49]. More genetic information on cross-reactive T cells will further elucidate metal allergy and facilitate safety in dermatology and dental treatments. This new mouse model is useful for the diagnosis of intraoral metal contact allergy, and the development of new treatments to metal-specific T cells in the oral mucosa.

These results suggest that metal allergy immune response in the oral mucosa differs from the skin. Ni exposure is believed to cause metal allergy [50]. Thus, immunological information in the cross-reactivity of metal allergy could be important for selecting dental materials to prevent incompatibility reactions.

## 4. Materials and Methods

### 4.1. Animals

Four-week-old female BALB/cAJcl mice (n = 56) were purchased from CLEA Japan (CLEA Japan, Tokyo, Japan) and housed under standard conditions. During the study period, all mice remained in good health, and they were assigned randomly to various groups. The mice were acclimated for at least 7 days before experimental use. They were kept in standard conditions (plastic and aluminum cages with a lid made of stainless-steel wire at our conventional animal facility that maintained the temperature at approximately 23 °C ± 1 °C and humidity at 30–70% with a 12 h day/night cycle). Food and water were available ad libitum.

### 4.2. Reagents

PdCl_2_ (>99% pure), NiCl_2_ (>99% pure), and CrCl_2_(>99% pure) were purchased from Wako Pure Chemical Industries ((FUJIFILM Wako Pure Chemical Co., Ltd., Osaka, Japan). Lipopolysaccharide (LPS) from Escherichia coli (O55:B5) prepared via phenol-water extraction was purchased from Sigma (Sigma-Aldrich, St. Louis, MO, USA). PdCl_2_, NiCl_2_, CrCl_2_, and LPS were dissolved in sterile saline (Otsuka Normal Saline, Otsuka Pharmaceutical Factory, Inc., Tokushima, Japan).

### 4.3. Anesthetic Agents

The following anesthetics were prepared: medetomidine hydrochloride (Nippon Zenyaku Kogyo Co., Ltd., Fukushima, Japan), midazolam (Sandoz, Tokyo, Japan), and butorphanol tartrate (Meiji Seika Pharma Co., Ltd., Tokyo, Japan). These anesthetic agents were kept at room temperature (RT).

Medetomidine hydrochloride was prepared at a dose of 0.3 mg/kg, midazolam at a dose of 4 mg/kg, and butorphanol tartrate at a dose of 5 mg/kg. The concentration ratio of the three types of mixed anesthetic agents was determined based on a previous study [51]. Therefore, 0.75 mL of medetomidine hydrochloride was mixed with 2 mL of midazolam and 2.50 mL of butorphanol tartrate and was adjusted to a volume of 19.75 mL with sterile saline. All agents were diluted in sterile saline and stored at 4 °C in the dark. The three types of mixed anesthetic agents were administered to all mice at a volume of 0.01 mL/g of body weight.

### 4.4. Experimental Protocol of the Metal Allergy Cross-Reaction Mouse Model

The protocols were used based on previous protocols for the induction of metal allergy in the oral mucosa [5]. Each experimental group of mice was separated into six sets, each comprising randomly chosen mice (Table 1). All experiments were carried out in another room after transfer from the animal holding room.

Sensitization: In total, 125 µL of 10 mm PdCl_2_, NiCl_2_, and CrCl_2_ and 10 μg/mL of LPS in sterile saline were injected twice at an interval of 7 days via the intradermal route into the left and right postauricular skin of mice (250 µL each). Seven days after the second sensitization, the mice were challenged for the first time.

Challenge for elicitation: At day 7 after the second sensitization, the ACM mice were challenged for elicitation with 25 µL of 10 mm PdCl_2_, NiCl_2_, and CrCl_2_ without LPS in sterile saline into the left and right buccal mucosa via submucosal injection under anesthesia with the three types of mixed anesthetic agents. The mice with metal allergy cross-reactions were classified into four groups: sensitization to NiCl_2_ with LPS and challenged with PdCl_2_ (Ni-Pd) group (n = 5), CrCl_2_ (Ni-Cr) group (n = 5), sensitization to CrCl_2_ with LPS and challenged with NiCl_2_ (Cr-Ni) group (n = 5), and sensitization to PdCl_2_ with LPS and challenged with NiCl_2_ (Pd-Ni) group (n = 5). Sensitization and challenge for elicitation used different metal solutions for each. Mice sensitized with NiCl_2_ plus LPS and then challenged with sterile saline were used as a control.

### 4.5. Measurement of Oral Mucosa Swelling

Buccal mucosa swelling was measured before challenge and at 24 h, 72 h, and 1 week after the first challenge using a Peacock dial thickness gage (Ozaki MFG Co., Ltd., Tokyo, Japan). The difference in oral mucosa thickness before and after challenge was recorded. All procedures were performed by the same experimenter.

### 4.6. Immunohistochemistry

Buccal mucosa specimens were obtained from mice with metal allergy cross-reaction ACM for histology and immunohistochemical analyses. Tissue samples were immersed in 4% paraformaldehyde–lysine–periodate overnight at 4 °C. After washing with phosphate-buffered saline (PBS) for 10 min, fixed tissues were penetrated by soaking in 5% sucrose/PBS for 1 h, 15% sucrose/PBS for 3 h, and then 30% sucrose/PBS overnight at 4 °C. Tissue samples were embedded in Tissue Mount (Chiba Medical, Saitama, Japan) and snap-frozen into a mixture of acetone and dry ice. Frozen sections were sliced into 6-µm-thick cryosections and air dried on poly-L-lysine-coated glass slides. For histological analyses, the cryosections were stained with H&E. Antigen retrieval was performed for immunohistochemical analyses. Cryosections were stained with anti-mouse F4/80 (1:1000; Cl-A3-1, Abcam, Cambridge, UK) and anti-CD3 (1:500; SP7, Abcam, Cambridge, UK) monoclonal antibodies (mAbs). Non-specific binding of mAbs was blocked via the incubation of sections in PBS containing 5% normal goat and rabbit serum, 0.025% Triton X-100 (FUJIFILM Wako Pure Chemical, Osaka, Japan), and 5% bovine serum albumin (Sigma, Aldrich St. Louis, MO, USA) for 30 min at RT. The sections were incubated with primary mAbs for 1 h at RT. After washing three times with PBS for 5 min, intrinsic peroxidase was quenched using 3% hydrogen peroxide (H_2_O_2_) in methanol. After soaking the sections in distilled water, they were washed twice and then incubated with a secondary antibody (biotinylated goat anti-hamster IgG or biotinylated rabbit anti-rat IgG) for 1 h at RT. After soaking the sections in distilled water, they were washed twice. Then, sections were incubated with a secondary antibody (biotinylated goat anti-hamster IgG antibody or biotinylated rabbit anti-rat IgG antibody) for 1 h at RT. After washing three times, the sections were incubated with Vectastain ABC Reagent (Vector Laboratories, Burlingame, CA, USA) for 30 min at RT, followed by 3,3-diaminobenzidine staining (0.06% diaminobenzidine and 0.03% H_2_O_2_ in 0.1 M Tris-HCl, pH 7.6; Wako Pure Chemicals Co., Ltd., Osaka, Japan). The tissue sections were counterstained with hematoxylin to visualize the cell of nuclei.

### 4.7. RNA Extraction and cDNA Synthesis

Fresh buccal mucosa tissue specimens were obtained from each mouse and immediately soaked in RNAlater RNA Stabilization Reagent (Qiagen, Hilden, Germany). Total RNA from the buccal mucosa tissue was extracted using the RNeasy Lipid Tissue Mini Kit (Qiagen) according to the manufacturer’s instructions. Complementary DNA (cDNA) was synthesized from DNA-free RNA using the PrimeScript™ RT reagent Kit (Takara Bio, Tokyo, Japan) according to the manufacturer’s instructions.

### 4.8. Quantitative Polymerase Chain Reaction

The expression levels of immune response-related genes, including T cell-related CD antigens, cytokines, cytotoxic granules, transcription factors of regulatory T cells, CD1d-restricted T cells, and MHC-related protein 1 were evaluated via quantitative polymerase chain reaction (qPCR) using the Bio-Rad CFX96 System (Bio-Rad, Hercules, CA, USA). Specific primers for GAPDH, CD4, CD8, IFN-γ, TNF-α, IL-4, IL-5, Foxp3, CD1d, MR1, and granzymes A and B have been described in previous studies [52,53]. Freshly isolated total RNA from the buccal mucosa tissue of mice was converted to cDNA using PrimeScript RT Reagent Kit (Takara Bio) according to the manufacturer’s instructions. The PCR comprised 5 µL of SsoFast™ EvaGreen^®^ Supermix (Bio-Rad), 3.5 µL of RNase/DNase-free water, 0.5 µL of 5-µM primer mix, and 1 µL of cDNA, with a final volume of 10 µL. The cycling conditions were as follows: 30 s at 95 °C, followed by 45 cycles of 1 s at 95 °C and 5 s at 60 °C. At the end of each cycle, melting curve analysis was performed from 65 °C to 95 °C to confirm the homogeneity of PCR products. All assays were repeated three times, and the mean values were calculated at the gene expression levels. Five 10-fold serial dilutions of each standard transcript were used to determine the absolute quantification, specification, and amplification efficiency of each primer set. Standard transcripts were generated by the in vitro transcription of the corresponding PCR product in a plasmid. The nucleotide sequences were confirmed via DNA sequencing using the CEQ8000 Genetic Analysis System (Beckman Coulter, Fullerton, CA, USA). Their quality and concentration were validated using the Agilent DNA 7500 Kit in an Agilent 2100 Bioanalyzer (Agilent, Santa Clara, CA, USA). The expression of the GAPDH gene was used as an internal control. The expression levels of each target gene were normalized to GAPDH expression.

### 4.9. Statistical Analysis

Differences between the mean values of each experimental group were analyzed using the Kruskal–Wallis test, followed by Dunn’s multiple comparison tests and the Mann–Whitney U-test using GraphPad Prism 7 software for Windows (GraphPad Software, Inc., San Diego, CA, USA). *p* value of <0.05 was considered significant; *p* value of <0.01, highly significant; and *p* value of <0.001, extremely significant.

## 5. Conclusions

The Ni-sensitized group showed significant differences in buccal mucosal swelling, and the expression of CD8, cytotoxic granules, and inflammation-related cytokines compared with the control and Pd-sensitized groups. Ni sensitization and Pd and Cr challenge can cause cross-reactivity in intraoral metal allergy.

## Figures and Tables

**Figure 1 ijms-24-03965-f001:**
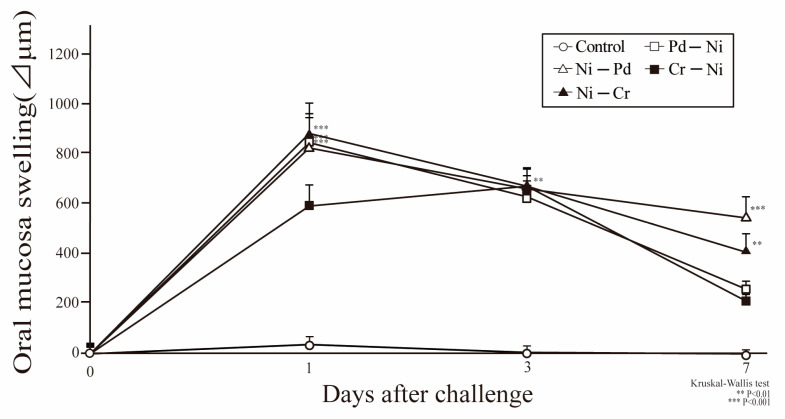
Oral mucosa swelling in mice with cross-reactive metal-induced allergy. In all groups, swelling was measured 1, 3, and 7 days after the first challenge. Furthermore, sensitization to Ni-, Pd-, and Cr-induced allergies was evaluated 7 days after the first challenge. Bars and error bars indicate the mean + standard deviation (SD). Statistical significance was evaluated using the Kruskal–Wallis test, followed by Dunn’s multiple comparison tests. ** *p* value of <0.01 was considered very significant, and *** *p* value of <0.001 was considered extremely significant.

**Figure 2 ijms-24-03965-f002:**
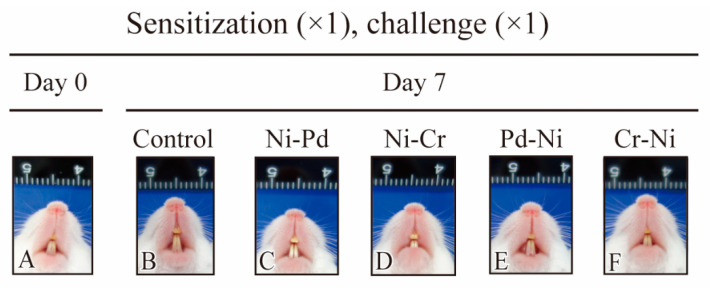
Swelling of the buccal mucosa of mice before challenge (**A**), and 7 days after the first challenge (**B**–**F**) were shown. Comparison of buccal mucosal swelling 7 days after the first challenge in control mice (**B**) and cross-reactive metal-induced allergic mice (**C**–**F**).

**Figure 3 ijms-24-03965-f003:**
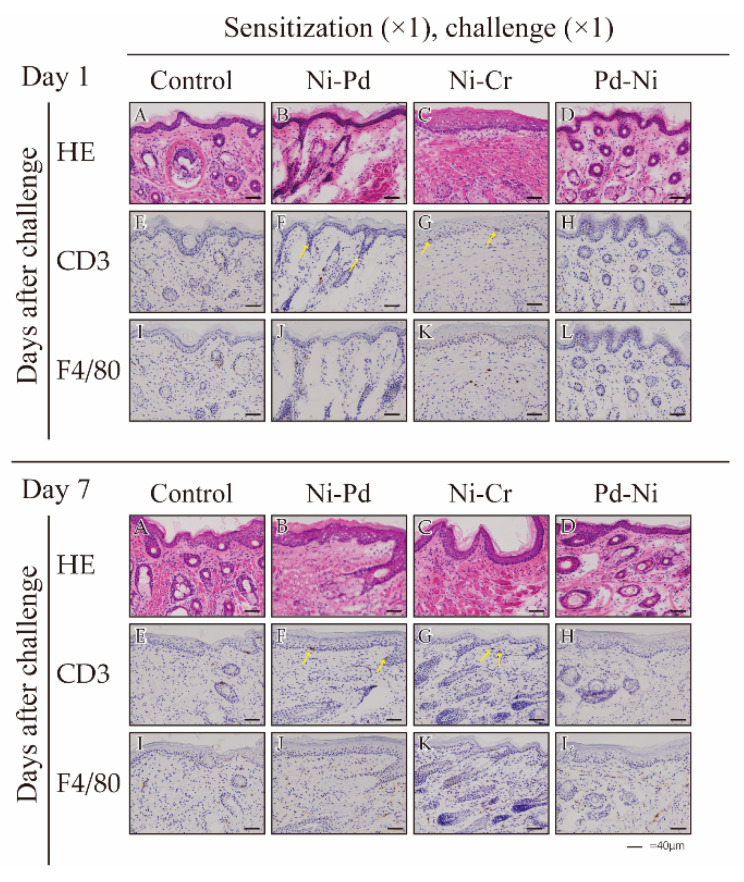
Histopathology and immunohistochemical analyses of accumulated T cells and antigen-presenting cells (APCs) in mice with cross-reactive metal induced allergy. Histopathology and immunohistochemical analyses of monoclonal antibody (mAb) that binds to a surface molecule on CD3-positive T cells and mature macrophages and dendritic cells (F4/80-positive cells) in buccal mucosa tissues. Frozen oral mucosa tissue sections were stained with hematoxylin and eosin (H&E) (**A**–**D**) and anti-CD3 (**E**–**H**) and anti-F4/80 (**I**–**L**) antibodies 1 and 7 days after the challenge. Representative examples of CD3-positive T cells are indicated by arrows. Scale bar = 40 µm.

**Figure 4 ijms-24-03965-f004:**
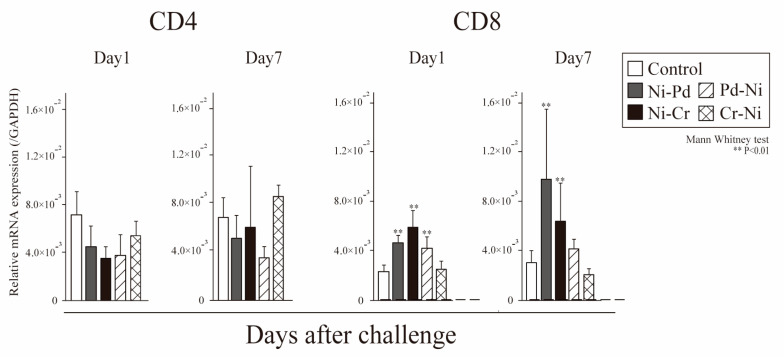
mRNA expression of T cell phenotypes in the oral mucosa of mice with cross-reactive metal-induced allergy. The mRNA expression of CD4 and CD8 in the buccal mucosa tissue was assessed 1 and 7 days after the first challenge. GAPDH gene expression was used as an internal control. Bars and error bars indicate the mean + standard deviation (SD). Statistical significance was tested using the unpaired Mann–Whitney test. ** *p* value of <0.01 was considered very significant.

**Figure 5 ijms-24-03965-f005:**
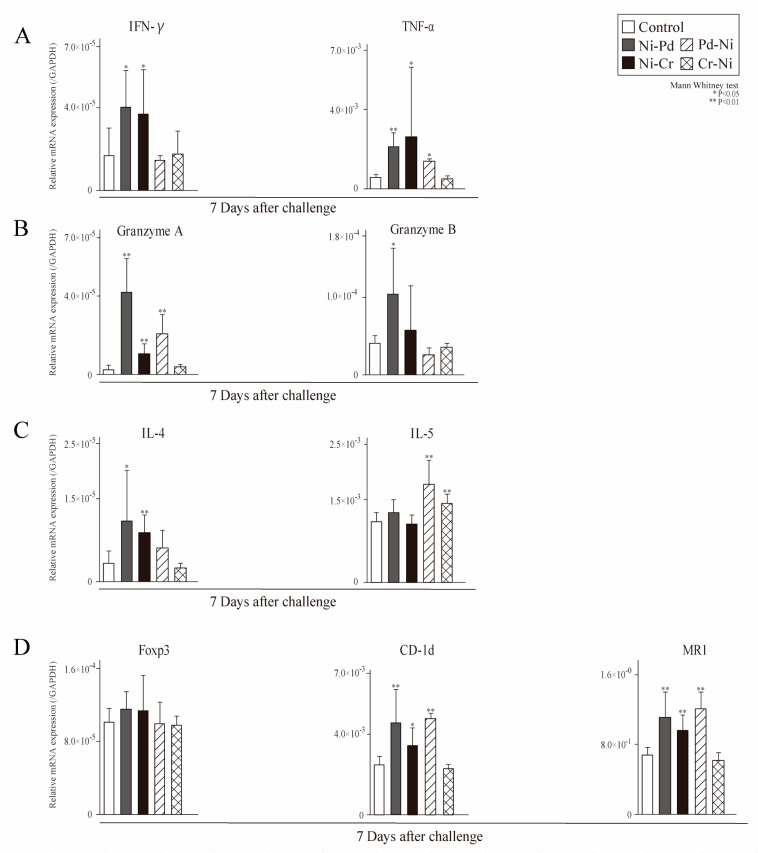
mRNA expression of T cell-related cytokines, cytotoxic granules, and T cell-related markers genes in the oral mucosa of cross-reactive metal-induced allergic mice. mRNA expression of (**A**) T helper type (Th) 1 cytokines (tumor necrosis factor(TNF)-α, Interferon(IFN)-γ), (**B**) cytotoxic granules (granzyme A and B), (**C**) T helper type (Th) 2 cytokines (IL-4 and IL-5), (**D**) transcription factors of regulatory T cells, CD1d-restricted T cells, and MHC-related protein 1 in the buccal mucosa tissue were assessed 7 days after the first challenge. GAPDH gene expression was used as an internal control. Bars and error bars indicate the mean + standard deviation (SD). Statistical significance was tested using the unpaired Mann–Whitney test. * *p* value of <0.05 was considered significant, and ** *p* value of <0.01 was considered very significant.

**Table 1 ijms-24-03965-t001:** Experimental groups of the metal allergy cross-reaction mouse model.

Groups ACM	Sensitization Metal Salts	Challenge for Elicitation Metal Salts
Ni *-Pd **	NiCl_2_	PdCl_2_
Ni-Cr ***	NiCl_2_	CrCl_2_
Pd-Ni	PdCl_2_	NiCl_2_
Cr-Ni	CrCl_2_	NiCl_2_

* Nickel; ** Palladium; *** Chromium.

## Data Availability

This data presented in this study are available on request from the corresponding author.

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
