# Peer review of "Cross-Reactivity of Intraoral Allergic Contact Mucositis in the Nickel-Sensitized Ear Model of Metal Allergy"

_ijms, 2023, doi:10.3390/ijms24043965_

Round 1
Reviewer 1 Report
Dear Authors, I have read with interest the manuscript as it highlights an important topic. In my opinion in the clinical setting we see more patients suffering from allergies or dermatitis. Cross-reactivity in intraoral metal allergy should be closely studied in order to improve dental materials selection to prevent allergic reactions in clinical settings. Firstly I would like to point out the need of restructuring in order to make the manuscript clear and easier to read. My suggestion would be to place the section Materials and Methods after the Introduction section. Results The statistical analysis is rigorously performed and the results are presented in a clear manner. Discussion As the study has practical importance, I would suggest that the bibliography should be completed with more references preferably from the last 5 years. Materials and methods The research is replicable and the methods are repeatable and presented in a clear manner.Author Response
Thank you for your comments. We agree your suggestion, however, in this journal, it would be more appropriate because the MDPI template consisted of Introduction, Results, Discussion, and Material and Methods. As you indicated, we have completed the reference list by adding more recent references from the last five years. Discussion descriptions have been added and modified.

Reviewer 2 Report
The current study aimed to establish a novel mouse model of Ni ear sensitization and characterize intraoral infiltrating T cells during the elicitation phase in terms of phenotypic T cell markers and cytokine expressions.
The study is well-written, and it reads well. I have only one comment:
The authors need to add a statement about the methodology in the abstract.
Author Response
Thank you for your high evaluation of this paper. As you indicated, the abstract needs to include a description of the methodology. Thank you for providing these insights. According to your comment, we changed the sentence in the Abstract section. Two sensitizations with nickel, palladium, and chromium plus lipopolysaccharide solution into the postauricular skin were followed by a single nickel, palladium, and chromium challenge of the oral mucosa to generate the intraoral metal contact allergy mouse model.

Reviewer 3 Report
1. How did the researchers indicate about the oral mucosa swelling? What was the indicator and how you determine the difference amongst the group?
2. Figure 1 showed the treatment group but, in the M & M yet unclear how the researchers mix between metals? How was the composition each mixture? Was it 50-50 for example for Ni-Cr, Ni-Pd etc?
3. Figure 2 still unclear what the authors wanted to show? It seemed that all the groups were giving almost the same features. Where was showing the redness and/or swelling and which part? The use of arrow or line will be suggested for this part?
4. Figure 3, was it taken from the representatieve data? It was also unclear what the authors wanted to show. Use arrow?
5. In line 312-314, Authors mentioned about the use of LPS but none of data were showed in the presence of LPS as well as in the disscussion section.
6. Cross reactivity was evaluated after challenged in 1 and 7 days, however there isn’t sufficient explanation in the discussion sections about the use of 2 different time points.
7. Based on the result of the study, what would researchers suggested for the implications clinically in dentistry?
8. The conclusion section should have the main results in quantitative statements as well.
Round 2
Reviewer 3 Report
The authors address most of my comments. Therefore, I recommend the manuscript for publication